# Study on the Source Profile Characteristics of Carbon Plant

**Sen Li [1], Danni Liang [2,*] and Jianhui Wu [3]**

1    Tongliao Ecological Environment Monitoring Station of Inner Mongolia Autonomous Region, Tongliao 028001, China; lisen6203600@gmail.com

2    Tianjin Shuangyun Environmental Protection Technology Co., Ltd., Tianjin 300350, China

3    State Environmental Protection Key Laboratory of Urban Ambient Air Particulate Matter Pollution Prevention and Control, College of Environmental Science and Engineering, Nankai University, Tianjin 300350, China; envwujh@nankai.edu.cn

*    Correspondence: liangdanni529@gmail.com; Tel.: +86-138-2173-5387

**Abstract:** In the background of carbon neutrality, carbon emissions are basked in the attention. As a significant source of carbon emissions, the emission characteristics of carbon plant should be known. Particulate matter in flue gas was collected in a carbon plant in Tongliao. The chemical components in $PM_{10}$ and $PM_{2.5}$ were analyzed, and source profile of carbon plant was established. The results showed that the mass fractions of EC, Ca, $Ca^{2+}$, S, Al, Si and Fe were higher in particles than other components. The chemical marker of carbon plant was EC, and the trace carbonaceous components of carbon plant were EC1 and EC2, which were very different from other carbon emission sources. In the absence of other chemical composition information, eight carbonaceous components can be used to identify the sources of particle.

**Keywords:** carbon plant; source profile; carbonaceous components; $PM_{2.5}$; $PM_{10}$





## 1. Introduction

As a supporting role in electrolytic aluminum industry, carbon plant has a history of more than 80 years in China, and has been put into operation in the northeast of China, Inner Mongolia and other regions [1]. In the process of production, carbon plant will emit a large amount of waste gas and particulate matter, that leads to atmosphere pollution to some extents [2,3].

Source profile is very important in source apportionment of particulate matter, especially in the process of receptor model allocation [4,5]. Source profile is the "fingerprint" of pollution sources, and can accurately define the emission characteristics of pollution sources [6]. Only by establishing true and representative source profile, the accuracy of source apportionment results of particulate matter can be ensured. The previous study on the pollution characteristics of carbon plants was mainly concerned with flue gas. Qin et al. [7] found that VOCs emitted by carbon plants in Zhengzhou were mainly aromatic hydrocarbons and OVOCs. Shao et al. [8] focused on the design of asphalt fume control system in carbon plants. Fang et al. [9] found that the airborne dust of carbon plant had mutagenicity, asphalt fume and flue gas which had influence on chromosome and cellular immunity. There were relatively few studies on the pollution characteristics of particulate matter emitted by carbon plant, and the understanding of source profile of particulate matter remains to be enhanced.

To investigate the chemical components characteristics of particles emitted by carbon plant and enrich the source profiles of industrial enterprise in China, particulate matter in flue gas was collected and analyzed for chemical components in this study. Source profiles of $PM_{10}$ and $PM_{2.5}$ emitted by carbon plant were established in support for the source apportionment of atmospheric particulate matter.

## 2. Materials and Methods

### 2.1. Sample Collection

Particulate matter in flue gas was collected in Tongliao Carbon Plant, which was in the suburb of Keerqin District. The information of sampling site and sampling method etc. was shown in Table 1.

**Table 1.** Information about sampling site.

| Information | Data |
| --- | --- |
| name of sampling site | Tongliao Carbon Plant |
| boiler type | coal-powder boiler |
| boiler tonnage | 8 t/h |
| desulfurization method | lime method |
| denitration method | none |
| dust removal method | cloth bag and filter cylinder |
| sampling method | dilution four-channel |
| dilution multiplication factor | 3 |
| sampling duration | 2 h 10 min |
| sampling flow | 33.34 L/min |
| number of samples | two $PM_{10}$, two $PM_{2.5}$ |

Particulate matter in flue gas was collected by dilution four-channel sampling instrument (PDSI-01P, Shanxi Zhengda Environmental Protection Technology Co., Ltd., Shanxi, China). $PM_{10}$ and $PM_{2.5}$ samples were collected simultaneously (two $PM_{10}$, two $PM_{2.5}$) by quartz filter membrane and polypropylene filter membrane (Pall Corporation, New York, NY, America). Preliminary experiment was conducted before actual sampling, with basic parameters of flue gas (flue gas temperature, humidity, etc.) well-monitored to determine the duration of sample collection. After preliminary experiment, $PM_{10}$ and $PM_{2.5}$ samples were collected in the chimney. The sample dilution multiplication factor is 3. After dilution, the temperature and relative humidity were 50 °C and 5%, respectively. The sampling duration was 2 h 10 min with a flow of 33.34 L/min and the total sampling volume was 1079 L.

### 2.2. Sample Analysis

#### 2.2.1. Blank Filter Membrane Treatment

Quartz filter membranes were baked at 600 °C for more than two hours in the Muffle oven, while polypropylene filter membrane was baked at 60 °C for more than two hours in the baking oven.

#### 2.2.2. Filter Membrane Weighing

The mass of particulate on the samples was determined gravimetrically by the filter membranes during pre and after post sampling period. The filter membranes were stored in an environment with constant temperature and humidity (20 ± 2 °C 40 ± 4%) for more than 48 h before weighing. One part in 100,000 electronic balance was used for weighing. Each filter was weighed at least three times until the difference between any two weighing results becomes less than 0.04 mg.

#### 2.2.3. Elemental Composition Analysis

1/4 of polypropylene filter membrane after sampling was cut into pieces and put into a microwave digestion tank, and then added with 3 mL $HNO_3$, 1 mL HCl, 1 mL $H_2O_2$ and 5 mL ultrapure water successively. Subsequently, the task was put into the microwave digestion instrument for digestion, following which the cooled solution was moved into a volumetric flask with volume set to 25 mL with ultrapure water. ICAP7400 inductively coupled plasma emission spectrometer was used for elements analysis (Al, As, Ca, Cr, Cu, Fe, K, Mg, Si, Zn, etc.).

### 2.2.4. Water-Soluble Ions Analysis

1/4 of quartz filter membrane after sampling was cut into pieces and put into a centrifuge tube, and then was added with 8 mL ultrapure water as well as ultrasonic to extract for 20 min. Next, the centrifugate was put into a refrigerator for 24 h, and a needle was used to drain the intermediate fluid. Lastly, the processed centrifugate was injected into an autosampler sample bottle filtered through a 0.2 μm filter head. The concentrations of $Na^+$, $NH_4^+$, $K^+$, $Mg^{2+}$, $Ca^{2+}$, $F^-$, $Cl^-$, $Br^-$, $SO_4^{2-}$ and $NO_3^-$ were analyzed by ICS-900 ion chromatograph with the detection limits of 0.019, 0.020, 0.025, 0.037, 0.020, 0.010, 0.012, 0.027, 0.027, 0.030 μg/m$^3$, successively.

### 2.2.5. Carbon Analysis

DRI2001A thermo-optic carbon analyzer was used for to analyze organic carbon (OC) and elemental carbon (EC). The detection limits of OC and EC were 0.29 μgC/cm$^2$ and 0.01 μgC/cm$^2$. The analysis of carbon components adopted the IMPROVE_A heating procedure [10,11]. OC1, OC2, OC3 and OC4 were measured at 140 °C, 280 °C, 480 °C and 580 °C in an anaerobic condition (100% helium). After that, EC1, EC2, EC3 were measured at 580 °C, 740 °C and 840 °C in an aerobic condition (98% helium, 2% oxygen). The content of optical pyrolyzed carbon (OP) was determined by irradiating the samples with a 633 nm He-Ne laser [10,11]. OC (Organic Carbon) and EC (Elemental Carbon) are defined as follows:

$$OC = OC1 + OC2 + OC3 + OC4 + OP \tag{1}$$

$$EC = EC1 + EC2 + EC3 - OP \tag{2}$$

Note that at least one set of laboratory and method gaps should be included in each batch of test sample analysis. Contamination or loss should be avoided in every step. Detailed, methods of this analysis are showed in references [12–14].

## 3. Results and Discussions

### 3.1. Source Profile Characteristics of Carbon Plant

The mass proportions of major chemical components in $PM_{10}$ and $PM_{2.5}$ were shown in Figure 1. The results showed that the mass proportions of major chemical components in $PM_{2.5}$ were EC, OC, $NO_3^-$, Ca, $Ca^{2+}$, S, Al, $Cl^-$, Si, Fe, Na and Mg, successively. In addition, the mass proportions of major chemical components in $PM_{10}$ were successively EC, Fe, Ca, OC, Al, S, Si, Mg, $NO_3^-$, $Ca^{2+}$, $Cl^-$ and $K^+$, successively.

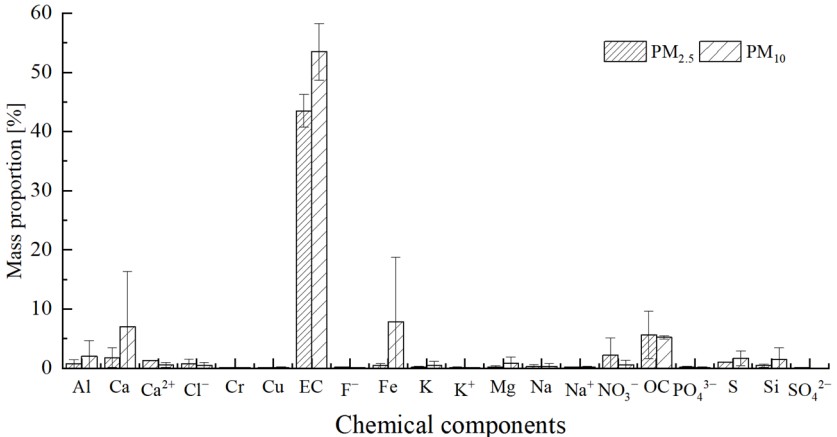

**Figure 1.** The mass proportions of major chemical components in $PM_{10}$ and $PM_{2.5}$ from carbon plant.

EC accounted for the highest proportion of particulate matter emitted by carbon plant, which were 43.491 ± 2.748% and 53.471 ± 4.824% in $PM_{2.5}$ and $PM_{10}$, successively. This result can be explained by the main products of the carbon plant which were carbon rods and graphite powder. The result of high mass proportions of Ca and $Ca^{2+}$ was due to

desulfurization by calcium method, and S may come from petroleum coke in the calcination stage and industrial natural gas in the calcination stage [15]. Al, Si, Fe and other crustal elements occupied a higher proportion in inorganic elements, and the mass fractions of these elements in $PM_{10}$ were higher than in $PM_{2.5}$, because crustal elements were primarily distributed on coarse particles [16]. The source profiles of $PM_{10}$ and $PM_{2.5}$ were shown in Table 2. It can be seen that the chemical maker of carbon plant is EC. The chemical maker of the source profile was also called the tracer species, which was referred as the component in a certain source. This chemical maker has a large impact on the source contribution value and the standard deviation of contribution value, and is considered as an important criterion to distinguish a certain source from other sources [17,18].

**Table 2.** The source profiles of $PM_{10}$ and $PM_{2.5}$ from carbon plant (%).

| Components | $PM_{2.5}$ | $PM_{10}$ |
| --- | --- | --- |
| Al | $0.766 \pm 0.723$ | $1.997 \pm 2.690$ |
| Ca | $1.796 \pm 1.661$ | $7.060 \pm 9.308$ |
| Co | $0.002 \pm 0.001$ | $0.001 \pm 0.001$ |
| Cr | $0.103 \pm 0.091$ | $0.123 \pm 0.002$ |
| Al | $0.766 \pm 0.723$ | $1.997 \pm 2.690$ |
| Cu | $0.085 \pm 0.057$ | $0.139 \pm 0.075$ |
| Fe | $0.450 \pm 0.383$ | $7.849 \pm 10.910$ |
| Hg | $0.004 \pm 0.005$ | $0.014 \pm 0.019$ |
| K | $0.175 \pm 0.193$ | $0.505 \pm 0.674$ |
| Mg | $0.202 \pm 0.0219$ | $0.829 \pm 1.130$ |
| Mn | $0.007 \pm 0.003$ | $0.055 \pm 0.073$ |
| Na | $0.307 \pm 0.341$ | $0.330 \pm 0.445$ |
| Ni | $0.047 \pm 0.051$ | $0.082 \pm 0.104$ |
| Pb | $0.013 \pm 0.009$ | $0.030 \pm 0.036$ |
| S | $1.064 \pm 0.033$ | $1.674 \pm 1.222$ |
| Si | $0.456 \pm 0.275$ | $1.523 \pm 1.942$ |
| Ti | $0.012 \pm 0.009$ | $0.045 \pm 0.059$ |
| V | $0.008 \pm 0.006$ | $0.018 \pm 0.012$ |
| Zn | $0.030 \pm 0.008$ | $0.089 \pm 0.104$ |
| OC | $5.624 \pm 4.017$ | $5.228 \pm 0.264$ |
| EC | $43.491 \pm 2.748$ | $53.471 \pm 4.824$ |
| $F^-$ | $0.159 \pm 0.086$ | $0.071 \pm 0.053$ |
| $Cl^-$ | $0.756 \pm 0.820$ | $0.505 \pm 0.493$ |
| $Br^-$ | $0.042 \pm 0.012$ | $0.014 \pm 0.002$ |
| $NO_3^-$ | $2.224 \pm 2.908$ | $0.621 \pm 0.745$ |
| $PO_4^{2-}$ | $0.184 \pm 0.180$ | $0.123 \pm 0.116$ |
| $SO_4^{2-}$ | $0.091 \pm 0.035$ | $0.046 \pm 0.021$ |
| $Na^+$ | $0.164 \pm 0.115$ | $0.176 \pm 0.192$ |
| $NH_4^+$ | $0.010 \pm 0.012$ | $0.003 \pm 0.004$ |
| $K^+$ | $0.146 \pm 0.104$ | $0.089 \pm 0.103$ |
| $Mg^{2+}$ | $0.045 \pm 0.034$ | $0.019 \pm 0.020$ |
| $Ca^{2+}$ | $1.315 \pm 0.037$ | $0.609 \pm 0.422$ |

*3.2. Distribution Characteristics of Carbonaceous Components*

The proportions of eight carbonaceous components in total carbon from carbon plant were presented in Figure 2. As shown, EC1 and EC2 are very high in total carbon, accounted for over 85% of total carbon. The mass fractions of EC1 and EC2 in $PM_{2.5}$ were 42.55% and 45.46%. As for $PM_{10}$, the mass fractions of EC1 and EC2 were 32.35% and 58.68%. OC1, OC2, OC3, OC4, EC1 and EC3 accounted for a higher proportion of total carbon in $PM_{10}$ than in $PM_{2.5}$. In $PM_{10}$, only EC2 accounted for a higher proportion of total carbon than in $PM_{2.5}$. No OP was detected in $PM_{2.5}$.

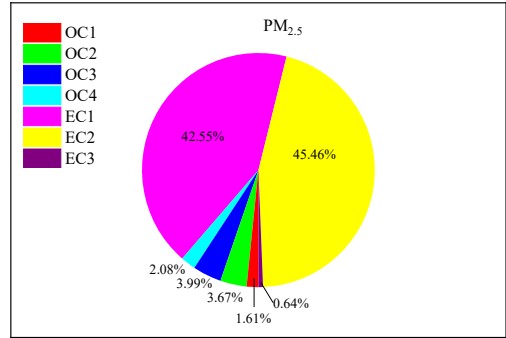
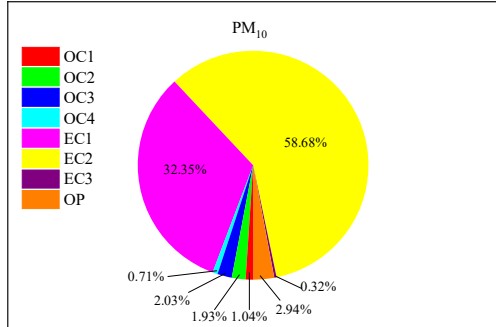

**Figure 2.** The mass fractions of eight carbonaceous components in total carbon of $PM_{10}$ and $PM_{2.5}$ from carbon plant.

### 3.3. Comparison with Other Emission Sources by Identifiable Carbonaceous Components

Based on the distribution characteristics of carbonaceous components as described above, the trace carbonaceous components of carbon plant were EC1 and EC2, which were significantly different from other emission sources. As shown in Table 3, the trace carbonaceous component of biomass burning was OC1. The trace carbonaceous components of coal combustion were OC1, OC2 and EC2. For different vehicles, EC2 was the trace carbonaceous component of construction machinery and diesel vehicle. As for motor vehicle, the trace carbonaceous components were OC1, OP, EC1, EC2 and EC3.

**Table 3.** Trace Carbon Components of Different Emission Sources.

| Emission Sources | Trace Carbonaceous Components | References |
| --- | --- | --- |
| carbon plant | EC1, EC2 | this study |
| biomass burning | OC1 | [19–22] |
| coal combustion | OC1, OC2, EC2 | [19,20,22] |
| motor vehicle | OC1, OP, EC1, EC2, EC3 | [19,20] |
| construction machinery | EC2 | [20] |
| catering industry | OC2, OC3 | [20,21] |
| diesel vehicle | EC2 | [23] |

The chemical components of particulate matter emissions were mostly affected by raw materials, combustion process, desulfurization facilities and ect. In the iron and steel industries in China, $SO_4^{2-}$, Al and $NH_4^+$ were the dominating components for the sintering source profiles. In addition, there was abundant Fe in pudding source profiles [24]. The content of OC and EC in coal charging was significantly higher than other components, which was largely affected by combustion process [25]. The OC, Al and Ca were relatively high in the cement kiln $PM_{2.5}$, while Al, $SO_4^{2-}$ and OC were relatively high in the coal-fired boiler $PM_{2.5}$ [26]. In this study, EC was the highest in $PM_{10}$ and $PM_{2.5}$ of carbon plant, because the major products of the carbon plant were carbon rods and graphite powder.

This study is aimed at enriching the source profiles of industrial enterprise in China. It was well-known that source profile was crucial to source apportionment of particulate matter, but emission sources sampling process was difficult because it was subject to industry environment and field conditions. On finite condition, this study only collected particulate matters samples from a carbon plant. For future studies, particle samples will be collected from various industrial enterprise to assess if carbonaceous components is able to distinguish different emission sources to the similar levels.

### 4. Conclusions

In order to investigate the emission characteristic of carbon plant, particles were collected in a carbon plant. The chemical components were analyzed in particles.

Key findings of this research are as follows:

- The percentages of EC, Ca, Ca$^{2+}$, S, Al, Si and Fe were higher in particles from carbon plant than the remaining components.
- The chemical marker of carbon plant was EC, and the trace carbonaceous components of carbon plant were EC1 and EC2, which were very different from other emission sources.
- In the absence of other chemical composition information, eight carbonaceous components can be used to identify the sources of particulate matter.

**Author Contributions:** Conceptualization, S.L. and D.L.; methodology, J.W.; software, J.W.; validation, S.L., D.L. and J.W.; formal analysis, D.L.; investigation, S.L.; resources, S.L.; data curation, D.L.; writing—original draft preparation, S.L.; writing—review and editing, D.L.; visualization, D.L.; supervision, J.W.; project administration, D.L. All authors have read and agreed to the published version of the manuscript.

**Funding:** This research received no external funding.

**Institutional Review Board Statement:** The study was conducted in accordance with the Declaration of Helsinki, and approved by the Institutional Review Board.

**Informed Consent Statement:** Informed consent was obtained from all subjects involved in the study.

**Data Availability Statement:** Not applicable.

**Conflicts of Interest:** The authors declare no conflict of interest.

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
