# Peer review of "Study on the Source Profile Characteristics of Carbon Plant"

_atmosphere, doi:10.3390/atmos13060969_

Round 1

Reviewer 1 Report

The authors studied emission profile of a carbon plant. Although, there are not many studies related to emissions from carbon plant and the topic can be important at least for carbon plant emission that have similar cleaning technology as the one mentioned in the paper, the manuscript should not be accepted to this journal if several major problems will not be corrected.

Major problems are summarized bellow.

  • Discussion of the results is really extremely short (one paragraph). It must be extended substantially. The authors can discuss more origin of various elements in emissions, relation to composition of raw materials etc.
  • Methodology description must be improved. More details about sampling place in studied technology, sampling procedures, temperatures, sample dilution, instruments and materials used should be provided including instrument producers, filter material origin, number of samples analyzed etc.
  • It is not clear what material is actually analyzed . Bottom ash or flue gas aerosol? The sampling method itself is suspicious especially with regard to PM2.5 and PM10 sampling of bottom ash. The sampled material is extremely difficult to resuspend back to previous size distribution and the authors do not provide any clue that would confirm their assumptions.

Minor problems not mentioned yet.

Line 96 – proportion of EC in PM2.5 in the text is higher than in PM10 while it is opposite in the Fig 1 and in Tab. 2.

Lines 112-118 – the whole paragraph should be placed in methodology section not in results.

English should be improved,  terminology is sometimes weird (e.g. "identified element of carbon plant")

Figures can be improved to be more readable.

There are more minor correction needed but they are mentioned in description of major problems in general way.

Author Response

Response to Reviewer 1Comments

Point 1:Discussion of the results is really extremely short (one paragraph). It must be extended substantially. The authors can discuss more origin of various elements in emissions, relation to composition of raw materials etc.

Response 1: Discussion of the results was expanded, The relationship between chemical components of particulate matter emissions and raw materials, combustion process, desulfurization facilities and so on were discussed.

Point 2:Methodology description must be improved. More details about sampling place in studied technology, sampling procedures, temperatures, sample dilution, instruments and materials used should be provided including instrument producers, filter material origin, number of samples analyzed etc.

Response 2: Methodology description was improved. Details about sampling site and sampling method were added in 2.1.

Point 3:It is not clear what material is actually analyzed . Bottom ash or flue gas aerosol? The sampling method itself is suspicious especially with regard to PM2.5 and PM10 sampling of bottom ash. The sampled material is extremely difficult to resuspend back to previous size distribution and the authors do not provide any clue that would confirm their assumptions.

Response 3: The analyzed chemical components was in flue gas aerosol. Bottom ash was collected for riching sample bank, while it was’t analyzed in this study. To avoid ambiguity, the description about bottom asd was deleted fully.

Point 4:Line 96 – proportion of EC in PM2.5 in the text is higher than in PM10 while it is opposite in the Fig 1 and in Tab. 2.

Response 4: Line 96 made a mistake, and it was revised in the new version of my paper.

Point 5:Lines 112-118 – the whole paragraph should be placed in methodology section not in results.

Response 5: Lines 112-118 – the whole paragraph was moved to the methodology section.

Point 6:English should be improved,  terminology is sometimes weird (e.g. "identified element of carbon plant")

Response 6: English was improved.

Point 7:Figures can be improved to be more readable.

Response 7: Figure 2 was modified to a color graph to increase readability. 

Point 8:There are more minor correction needed but they are mentioned in description of major problems in general way.

Response 8: The paper was checked and revised fully.

Reviewer 2 Report

In the introduction "apportionment" is used severl times, I suggest to use also some synosimous (e.g., distribution, allocation...).

Line 30: "Only by establishing true and representative source profile can we ensure" should be: Only by establishing true and representative source profile we can  ensure 

Line 119: check the bracket

Author Response

Response to Reviewer 2Comments

Point 1:In the introduction "apportionment" is used severl times, I suggest to use also some synosimous (e.g., distribution, allocation...).

Response 1: In Line 29 , apportionment was replaced by allocation.

Point 2:Line 30: "Only by establishing true and representative source profile can we ensure" should be: Only by establishing true and representative source profile we can  ensure 

Response 2: Line30"Only by establishing true and representative source profile can we ensure" was revised to:”Only by establishing true and representative source profile we can  ensure “

Point 3:Line 119: check the bracket

Response 3: All the brackets were checked to make sure they were conformed to format requirements.

Reviewer 3 Report

This paper studies the characteristics of source profile in carbon plant, distribution in carbonaceous components, and comparison with other emission sources. In addition, this paper provides meaningful insights and research for carbon plant emission characteristics.

While the subject of the analysis is very meaningful and the content of the article is very abundant, the manuscript needs to be revised before accepted for publication. My detailed comments are as follows:

  1. In the abstract, it is suggested adding background information on carbon emissions.
  2. Please keep the format of the tables in this paper, for example refer to Table 3. In particular, the format of Table 2 is different from that of Table 3.
  3. I think the content in Table 1 is a little monotonous. It is suggested to add additional sampling information in Table 1.
  4. It is suggested to appropriately increase the information expressed in the figure to make the content of article look richer.
  5. It is suggested that different colors can be used to fill the pie chart in Figure 2, which will make it easier for readers to understand.
  6. It is recommended to re-edit the format of the formula numbers in the article, and pay attention to the format requirements.
  7. The article should be revised in terms of language usage. It is recommended to check the language of the article.

Author Response

Response to Reviewer 3Comments

Point 1:In the abstract, it is suggested adding background information on carbon emissions.

Response 1:Background information on carbon emissions was added in the abstract.

Point 2:Please keep the format of the tables in this paper, for example refer to Table 3. In particular, the format of Table 2 is different from that of Table 3.

Response 2:The format of Table 2 was modified.

Point 3:I think the content in Table 1 is a little monotonous. It is suggested to add additional sampling information in Table 1.

Response 3: Sampling information was added in Table1.

Point 4:It is suggested to appropriately increase the information expressed in the figure to make the content of article look richer.

Response 4: The description of Figure2 was expaned.

Point 5:It is suggested that different colors can be used to fill the pie chart in Figure 2, which will make it easier for readers to understand.

Response 5: Figure 2 was changed to a color graph.

Point 6:It is recommended to re-edit the format of the formula numbers in the article, and pay attention to the format requirements.

Response 6: The format of the formula was modified to satisfy format requirements.

Point 7:The article should be revised in terms of language usage. It is recommended to check the language of the article.

Response 7:The language usage was revised.

Round 2

Reviewer 1 Report

The text was improved, but still moderate change is needed and many minor. I suggest to join Result and Discussion sections, to avoid very short Discussion part. Part of discussion is actually done in Results (lines 114-126, 141-150).

Minor changes are related mainly to English language, that needs extensive editing by native speaker. 

Examples of language mistakes:

Use the word "characteristic" or similar instead of "identifiable"  or "identified" that do not make any sense.

line 57 - "..should be monitored"  "were monitored" is just enough

lines 77, 78  - Sentences are written in imperative, it must be changed.

many similar language errors are present.

Another minor problems:

Line 55 - It is not clear what is meant by "presampling"

Line 72-73 - weighing procedure description must be improved, it is not clear what they mean by "loop weighing"  and also the rest of description is unclear.

Line 97 - OP determination is not described correctly, be more accurate or use a reference to the procedure.

Numbering of chapters from Results must be changed.

Author Response

Response to Reviewer 1 Comments

Point 1: The text was improved, but still moderate change is needed and many minor. I suggest to join Result and Discussion sections, to avoid very short Discussion part. Part of discussion is actually done in Results (lines 114-126, 141-150).

Response 1:Result and Discussion sections were added to replace Rusults part and Discussion part.

Point 2: Minor changes are related mainly to English language, that needs extensive editing by native speaker.

Response 2: This paper was editied by a native speaker to avoid language mistakes.

Point 3: Use the word "characteristic" or similar instead of "identifiable"  or "identified" that do not make any sense.

Response 3: The word”identifiable component” was replaced by “chemical marker” or “tracer species”.

Point 4: line 57 - "..should be monitored"  "were monitored" is just enough

Response 4: line 57 - "..should be monitored" was replaced by "were monitored".

Point 5: lines 77, 78  - Sentences are written in imperative, it must be changed.

Response 5: lines 77, 78  - Sentences were changed.

Point 6: many similar language errors are present.

Response 6: Language errors were modified by a native speaker.

Point 7: Line 55 - It is not clear what is meant by "presampling"

Response 7:  Line 55 -"presampling" was replaced by”preliminary experiment ”.

Point 8: Line 72-73 - weighing procedure description must be improved, it is not clear what they mean by "loop weighing"  and also the rest of description is unclear.

Response 8: Line 72-73 - weighing procedure description was improved.

Point 9: Line 97 - OP determination is not described correctly, be more accurate or use a reference to the procedure.

Response 9: References was added to describe OP.

Point 10: Numbering of chapters from Results must be changed.

Response 10: Numbering mistakes were modified.